# Improving yield and fruit quality traits in sweet passion fruit: Evidence for genotype by environment interaction and selection of promising genotypes

**Lourdes Maria Chavarría-Perez**[1], **Willian Giordani**[1], **Kaio Olimpio Graças Dias**[1], **Zirlane Portugal Costa**[1], **Carolina Albuquerque Massena Ribeiro**[1], **Anderson Roberto Benedetti**[1], **Luiz Augusto Cauz-Santos**[1], **Guilherme Silva Pereira**[2], **João Ricardo Bachega Feijó Rosa**[1], **Antonio Augusto Franco Garcia**[1], **Maria Lucia Carneiro Vieira**[1] *

1 Departamento de Genética, Escola Superior de Agricultura "Luiz de Queiroz", Universidade de São Paulo, Piracicaba, Brasil, 2 Bioinformatics Research Center, North Carolina State University, Raleigh, North Carolina, United States of America

* mlcvieir@usp.br

**Data Availability Statement:** All relevant data are within the paper or in its Supporting Information files.

## Abstract

Breeding for yield and fruit quality traits in passion fruits is complex due to the polygenic nature of these traits and the existence of genetic correlations among them. Therefore, studies focused on crop management practices and breeding using modern quantitative genetic approaches are still needed, especially for *Passiflora alata*, an understudied crop, popularly known as the sweet passion fruit. It is highly appreciated for its typical aroma and flavor characteristics. In this study, we aimed to reevaluate 30 genotypes previously selected for fruit quality from a 100 full-sib sweet passion fruit progeny in three environments, with a view to estimating the heritability and genetic correlations, and investigating the GEI and response to selection for nine fruit traits (weight, diameter and length of the fruit; thickness and weight of skin; weight and yield of fruit pulp; soluble solids, and yield). Pairwise genetic correlations among the fruit traits showed mostly intermediate to high values, especially those associated with fruit size and shape. Different genotype rankings were obtained regarding the predicted genetic values of weight of skin, thickness of skin and weight of pulp in each environment. Finally, we used a multiplicative selection index to select simultaneously for weight of pulp and against fruit skin thickness and weight. The response to selection was positive for all traits except soluble solids, and the 20% superior (six) genotypes were ranked. Based on the assumption that incompatibility mechanisms exist in *P. alata*, the selected genotypes were intercrossed in a complete diallel mating scheme. It is worth noting that all genotypes produced fruits, which is essential to guarantee yields in commercial orchards.

**Funding:** This work was supported by the following Brazilian institutions: Fundação de Amparo à Pesquisa do Estado de São Paulo (FAPESP, grant no. 2007/52607-5, and the scholarship awarded to LP, grant no. 2012/09100-5); Conselho Nacional de Desenvolvimento Científico e Tecnológico (CNPq, scholarships awarded to ZC, CR, and research grants awarded to MV and AG); and Coordenação de Aperfeiçoamento de Pessoal de Nível Superior (CAPES, Finance Code 001, and the scholarship awarded to AB). WG, KD, LC-S and JR were also supported by FAPESP scholarships (2018/09069-7, 16/12977-7, 2017/04216-9 and 2010/06702-9, respectively).

**Competing interests:** The authors have declared that no competing interests exist.

## Introduction

Brazil is the world's third largest producer of fruits, after China and India, and produces over 300 species. The most important crops are *Citrus* fruits, banana and pineapple (67% of total fruit production), followed by watermelon, coconut, papaya, grapes, apple and mango. Some native Brazilian fruit species are understudied, including the Brazilian guava (*Psidium guineense*), cashew (*Anacardium occidentale*) and passion fruits or passionflowers (*Passiflora* spp.).

Traditionally, passionflowers have been used as ornamental and medicinal plants, but they are primarily marketed as fresh fruit for immediate consumption and industrialized juice production. In Brazil, in particular, commercial crops are almost entirely based on a single native species, the sour passion fruit (*Passiflora edulis*), occupying around 90% of all orchards (see [1]). Its edible and aromatic fruits are used in juice concentrate blends consumed worldwide. A second species, the sweet passion fruit (*P. alata*), is native to the Brazilian plateau and the eastern Amazon region, but is cultivated as a low-intensity crop only in the South and Southeast of Brazil. It is appreciated for its typical aroma and flavor characteristics and can therefore command up to triple the price of the yellow passion fruit at local markets. Both crops provide a good alternative source of employment and income to small farmers.

*P. alata* is a semi-woody perennial climbing vine that produces large attractive, hermaphrodite flowers. In common with *P. edulis*, it is a diploid ($2n$ = 18), outcrossing, self-incompatible species [2–4]. For this reason, commercial orchards depend on visits by large, solitary native bees (*Xylocopa* spp.) or hand pollination, which is labor intensive and increases production costs.

A recent Brazilian survey on agricultural production showed that an area of 41,216 ha is planted with passion fruit, yielding 554,598 tonnes of fruit [5]. This is because fruit production is not stable and there is lack of improved varieties that meet the needs of both producers and consumers in terms of quality and yield. The production of sweet passion fruit relies on a few indoor selections, hampering large-scale farming. In 2017, the Brazilian Agricultural Research Corporation released the first sweet passion fruit cultivar for cropping mostly in the central region of the country for which it was developed (http://sistemas.agricultura.gov.br/snpc/cultivarweb/cultivares_registradas.php).

Although some work has been done on genetic and phenotypic analysis [6,7], studies focused on crop management practices and breeding using modern quantitative genetic approaches are still needed. In this context, understanding the genotype by environment interaction (GEI) has been one of the most important challenges faced by plant breeders, affecting every aspect of the decision-making process in breeding programs [8].

Many approaches can be adopted for modeling, analyzing and interpreting GEI in multi-environment trials (MET), including linear mixed models [9]. These models cover both fixed and random effects, and different variance–covariance (VCOV) structures can be used to explore random effects, such as heteroscedasticity and correlations [10]. For MET, these VCOV structures are usually modeled to investigate genotypic and residual correlations between environments. Additionally, genotype observations can be grouped according to levels of grouping factors, such as environments, providing a good representation of GEI [11]. Mixed models can be very useful for dealing with incomplete and unbalanced data from field experiments, such as data on fruit crops, and for estimating genetic parameters (heritability, genetic correlations, etc.). In this case, the preferred approach is REML/BLUP (Restricted Maximum Likelihood/Best Linear Unbiased Predictor) allowing genetic parameters to be estimated simultaneously and facilitating the prediction of genotypic values maximizing the correlation between true and predicted genotypic values [12].

With the aim of estimating the genetic and phenotypic parameters related to fruit traits and identifying the quantitative trait loci (QTLs) underlying these traits, our group has already researched a sweet passion fruit population consisting of 100 full-sibs from which 30 superior genotypes were selected [6]. In this study, we reevaluate these 30 genotypes in three environments, with a view to estimating the heritability coefficient and genetic correlations using VCOV matrices, investigating the GEI and predicting genetic values for fruit traits. As a consequence of our results, we have identified six superior genotypes, which were subsequently intercrossed in all combinations, including the reciprocals. Based on the assumption that incompatibility mechanisms exist in *P. alata* [13], the fruit set capacity in all pollinations has also become a goal of our study.

## Materials and methods

### Plant material

In this study, we examined 32 genotypes, consisting of a sample (n = 30) of full-sib progeny of sweet passion fruit and the two parents. Both parents are outbred and divergent accessions. The male parent, denoted SV3, was an indoor selection cultivated in the Southeast of Brazil (22˚17′ S, 51˚23′ W). The female parent, denoted 2(12), belongs to the progeny of a wild accession collected in a region between the Amazon and Cerrado ecosystems (15˚13′ S, 59˚20′ W); for details, see [4]. The SV3 accession is vigorous, develops faster and has vegetative organs larger than those of accession 2(12). It produces medium-sized to small egg-shaped fruits, and abundant aromatic pulp of a deep orange color. Accession 2(12) produces rounder, larger fruits with a thicker skin and less pulp that is a paler color. The 30 full-sibs were part of a progeny of 100 individuals previously evaluated in two environments over two growing seasons (see [6]).

### Field sites and measurements

The field studies did not involve endangered or protected species. The plant accessions are registered at Sistema Nacional de Gestão do Patrimônio Genético e do Conhecimento Tradicional Associado (SISGEN, Brazil, Registration no. A3FAE44). Field experiments were conducted at two locations and during two seasons, consolidated for a total of three environments (A, B and C) for the purpose of this study. Environments A and B were represented by seasons: A was conducted from January 2014 to August 2015 (1st season) and B from October 2015 to August 2016 (2nd season), both at the same location (Anhumas, SP, 22˚47' S, 48˚07' W, 500 m above sea level), while Environment C was represented by a 2nd season at a different location (Piracicaba, SP, 22˚42' S, 47˚38' W, 546 m above sea level). Both sites are in the Southeast of Brazil. All crop management practices were performed throughout the entire agricultural cycle. A randomized complete block design with six (environment A) or three (environments B and C) replicates was used, with the blocks arranged according to the field slope with three plants per plot arranged in rows. Plant and row spacing was 5 m (A) and 3 m (B and C). Plants were tied to 2-meter high wire trellises.

At the fruit set stage, up to 10 fruits per plant were harvested every week when the skin turned from green to yellow. The 30 fruits from each plot were then used to evaluate nine fruit traits: weight (WF, in g), diameter at the widest lateral point (DF, in mm); length of the fruit (LF, in mm); thickness of skin at the widest point (TS, in mm); weight of skin (WS, in g); weight (WP, in g) and yield of fruit pulp (YP, estimated as the quotient between WP and WF); soluble solids (SS, in ˚Brix), and yield (tonnes per ha). DF, LF and TS were measured using a stainless 0–200 mm digital caliper, and WF and WS using a digital scale (MARK 13000, Tecnal, Piracicaba, SP, Brazil). WP was calculated by subtracting WS from WF, and SS measured

using a portable sucrose refractometer with 0–32 °Brix scale (RTA-50, Instrutherm, SP, Brazil). In addition, the number of fruits produced per plant was noted at three different times prior to harvesting, and counting only those fruits present at about 3 weeks after blooming. Fruit production per plant (in kg) was calculated by multiplying the average number of fruits per plant by the mean fruit weight for the respective genotype. Finally, individual plant production was extrapolated on a per hectare basis as a function of the number of plants per hectare for estimating yield in tonnes per hectare.

## Statistical analysis

Single- and multi-environment trials analysis were fitted to linear mixed models in order to estimate the generalized measurement of heritability, find the adjusted means to obtain genetic correlations among traits, and rank genotypes for selection. Single-environment analyses were fitted for each trait using the following linear model:

$$y = \mu 1 + \mathbf{X}\mathbf{b} + \mathbf{Z}\mathbf{g} + \mathbf{e} \tag{1}$$

Where: $\boldsymbol{y}^{n \times 1}$ is the phenotype vector, related to $m$ genotypes and $j$ blocks; $n$ is the number of plots; $\mu$ is an intercept; $\mathbf{b}^{j \times 1}$ is the vector of block fixed effects, with $\mathbf{b} \sim \mathrm{MVN}(0, \sigma_b^2 \mathbf{I}_j)$, where MVN is a multivariate normal distribution; for genotypes, $\mathbf{g}^{m \times 1}$ is the vector of genotype random effects with $\mathbf{g} \sim \mathrm{MVN}(0, \sigma_g^2 \mathbf{I}_m)$; and $\mathbf{e}^{n \times 1}$ is the vector of the random effects of residuals with $\mathbf{e} \sim \mathrm{MVN}(0, \sigma_e^2 \mathbf{I}_n)$. The matrices $\mathbf{X}^{n \times j}$ and $\mathbf{Z}^{n \times m}$ represent the incidence of the respective fixed and random effects; $\mathbf{1}^{n \times 1}$ is a vector of ones; $\mathbf{I}_j$, $\mathbf{I}_m$ and $\mathbf{I}_n$ are identity matrices for the corresponding orders.

Multi-environment trial analyses were fitted using the following model:

$$y = \mu 1 + \mathbf{X}_1 \mathbf{b} + \mathbf{X}_2 \mathbf{r} + \mathbf{Z}_1 \mathbf{g} + \mathbf{e} \tag{2}$$

Where: $\boldsymbol{y}^{n \times 1}$ is the phenotype vector, related to $m$ genotypes, $q$ environments, and $j$ blocks; $n_i$ is the number of plots in each trial and $n$ is the total number of plots; $\mu$ is an intercept; $\mathbf{b}^{jq \times 1}$ is the vector of the fixed effects of the block within the environment; $\mathbf{r}^{q \times 1}$ is the vector of the environmental fixed effects, with $\mathbf{r} \sim \mathrm{MVN}(0, \sigma_r^2 \mathbf{I}_q)$, where MVN is a multivariate normal distribution; $\mathbf{g}^{m \times 1}$ is the vector of genotype random effects with $\mathbf{g} \sim \mathrm{MVN}(0, \Sigma g)$, where $\Sigma_g^{m \times q}$ is a genetic VCOV matrix and $\Sigma g = \boldsymbol{G}_m \otimes \boldsymbol{I}_q$; and $\mathbf{e}^{n \times 1}$ the vector of the random effects of residuals with $\mathbf{e} \sim \mathrm{MVN}(0, \Sigma e)$, where $\Sigma_r^{jq \times m}$ is a genetic VCOV matrix and $\Sigma e = \mathbf{I}_n \otimes \mathbf{R}$. The matrices $\mathbf{X}_1^{n \times q}$, $\mathbf{X}_2^{n \times jq}$ and $\mathbf{Z}_1^{n \times m}$ represent the incidence of the respective fixed or random effects; $\mathbf{1}^{n \times 1}$ is a vector of ones; $\mathbf{I}_{n_i}$, $\mathbf{I}_m$ and $\mathbf{I}_n$ are identity matrices for the corresponding orders.

The LRTs (Likehood Ratio Tests) were performed for single and multi-environment trial analyses for each trait. For MET, LRT values were obtained based on a model presenting a genetic effect and a GEI effect (interaction model). This interaction model is equivalent to a nested model containing a single term for the genotype effect nested within the environment, with a VCOV matrix based on the compound symmetry structure [19]. Therefore, compared to the interaction models, the nested GEI models have the advantage of evaluating different VCOV matrix structures which could be considered more realistic models [10, 33].

Linear mixed-model analysis was performed using the ASReml-R package [14] in which different VCOV structures were investigated for **G**, a matrix of random effects, and **R**, a matrix of residuals. A total of eight VCOV structures for random genetic effects (ID: identity; DIAG: diagonal; CShom: homogeneous compound symmetry; Ar1: first order autoregressive; Ar1H: heterogeneous first order autoregressive; CSHet: heterogeneous compound symmetry; UNST: unstructured; FA1: first order factor analysis) and two structures for **R** matrix residual effects

(identity and diagonal matrix) were tested and selected according to Akaike Information Criteria (AIC; [15]) and Bayesian Information Criteria (BIC; [16]). These criteria allow comparison of models with different random terms or VCOV structures, with a common fixed part, and evaluate the goodness-of-fit of each model, even if they are not nested. Due to the unbalanced situation, Residual Maximum Likelihood (REML; [17]) was applied for each trait. Based on the most appropriate models, the following estimates were then found: $\hat{\sigma}^2$ (residual variance), $\hat{\sigma}_f^2$ (phenotypic variance), $\hat{\sigma}_g^2$ (genetic variance), $\hat{\sigma}_e^2$ (environmental variance), $\hat{\sigma}_{ge}^2$ (GEI variance).

Genotypic correlations were estimated between environments by the expression $\frac{r_{k,k\prime}=\hat{\sigma}_{k,k\prime}}{\sqrt{(\hat{\sigma}_k^2 \ \hat{\sigma}_{k\prime}^2)}}$.

The BLUP values obtained from the selected model were used to compute the genotypic correlations using the Pearson's coefficient. The R package *psych* [18] was used to produce diagrams of dispersion between pairs of traits and plot the correlation networks.

Heritability was estimated using the approach in [19] which proposes an alternative expression when working with unbalanced data and mixed models:

$$H_c^2 = 1 - \frac{vBLUP}{2\sigma_g^2} \tag{3}$$

where: $H_c^2$ is the heritability value for each trait; $vBLUP$ variance is the average of the difference of two BLUPs, and $\hat{\sigma}_g^2$ is the genetic variance. The coefficient of genetic variation ($CV_g$) was calculated as $CV_g = \frac{\sigma_g}{\bar{x}}$ . 100, where $\hat{\sigma}_g$ is the square root of the estimated genetic variance and $\bar{x}$ the average for the population.

In order to select the superior 20% of full-sibs with the desired traits, four scenarios were created: the first three were based on single traits: selecting against TS; selecting against WS; and selecting for WP. In a fourth scenario, selection was based on a multiplicative selection index (MSI). The MSI was applied with the aim of increasing WP and decreasing TS and WS. It was calculated following the procedure proposed by [20]: for each individual $i$, $MI_i = \prod_{t=1}^{T}(\hat{y}_{it} - \lambda_t)$ was found, where $\hat{y}_{it}$ is the average adjusted value of the $i^{th}$ genotype for the $t^{th}$ trait, and $\lambda_t$ is the lower limit value accepted for the $t^{th}$ trait.

Thus, we determined:

$$RS = {}_{BLUPs} \tag{4}$$

$$RS(\%) = \frac{RS}{\bar{\bar{x}}_0} \ . \ 100 \tag{5}$$

where: $RS$ is the response to selection given by the BLUP values for the selected individuals ($\underline{BLUP_s}$); $\bar{x}_0$ is the phenotypic average for the sample and $RS(\%)$ is the percentage response to selection.

Finally, in order to investigate GEI, the GGE biplot (Genotype main effects + Genotype by environment interaction) [21] was generated using the *GGEBiplotGUI* software implemented in R.

## Reproductive compatibility between selected genotypes

Due to the potential for reproductive self-incompatibility in *P. alata*, which does not allow self-fertilization or fertilization involving genetically related individuals, the selected genotypes were crossed in a 6 × 6 diallel design, and the fruit set capacity evaluated at the same site (C), during the 2017/18 and 2018/19 growing seasons.

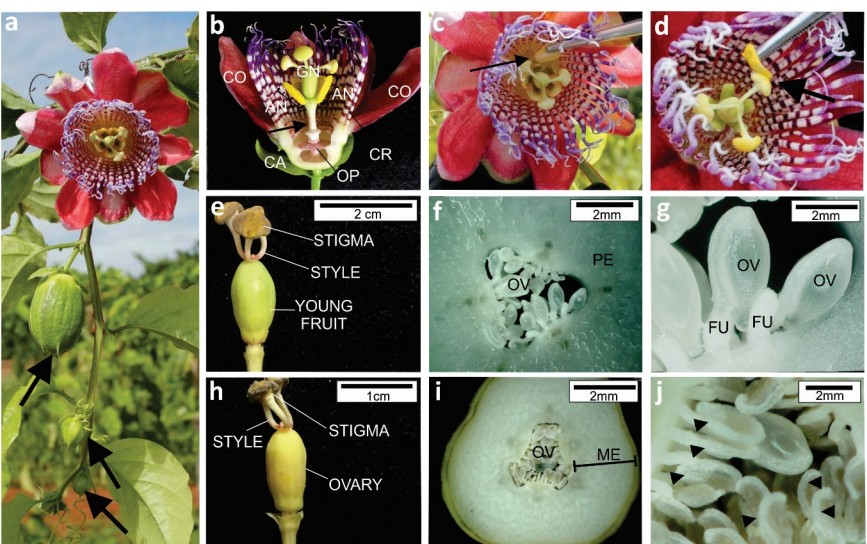

**Fig 1. Results from compatible and incompatible crosses in sweet passion fruit.** An open flower and closed buds (arrowed) (a); Flower structure, including calyx (CA), corolla (CO), androecium (AN), gynoecium (GN), corona (CR) and operculum (OP) (b); Steps involved in emasculating (c) and pollinating flowers (d); External morphology of compatible (e) and incompatible (h) crosses five days after pollination; Equatorial transversal section of the ovary of compatible (f, g) and incompatible (i, j) crosses observed under a digital microscope (Hirox KH-8700). Young fruit formation in a compatible cross (e), showing the pericarp (PE) (f) and turgid ovules (OV) attached to funicle (FU) (g). Chlorotic aspect and initiation of the ovary atrophy in an incompatible cross (h). Note that the mesocarp (ME) does not develop into a pericarp (j) and ovules are atrophied (j).

The plants were manually pollinated (Fig 1a–1j) using a procedure similar to that described by [22]. One day before anthesis, the flowers were protected with paper bags to avoid contamination. At the beginning of anthesis, the flowers were carefully emasculated using fine forceps and the anthers collected (Fig 1c and 1d). Pollination was performed by rubbing the anthers from one parent against the stigma of the other (Fig 1d). The flowers were then labeled and covered again with paper bags for 24 hours. Seven days after pollination, those flowers that had initiated fruit development were considered fertilized. Finally, the fruits were protected with a nylon bag to prevent falling during ripening. At least 10 stigmas were hand-pollinated for each cross, including reciprocal crosses and self-pollinations. The crosses were considered compatible (+) when > 50% of the pollinated flowers set fruits (Fig 1e–1g) and incompatible (−) in the absence of fruit set (Fig 1h–1j). Otherwise they were considered partially compatible.

## Results

### Single- and multi-environment trial analysis

Before carrying out the MET analysis, the phenotypic data from the single environment trials were analyzed. The likelihood ratio test, performed for each of the three environments, detected significant differences among genotypes for all the traits studied, revealing the levels of genetic variability within the population herein evaluated (Table 1).

To perform MET analysis, several VCOV structures for the **G** and **R** matrices were investigated and compared via AIC and BIC, attempting to find the most appropriate model for each trait (S1 Table), given that lower AIC and BIC values indicate a better fit. For WF, LF and WS, the best model was found by fitting an UNST VCOV matrix (heteroscedasticity and different genetic covariances) to the genotype effects, and an ID VCOV matrix (homogeneous variance

**Table 1. Likelihood Ratio Test (LRT) for Genotype and Genotype-By-Environment Effects (GEI) with regard to nine fruit traits according to single (A, B And C) and multi-environment trials.**

| Trait | Environment A | Environment B | Environment C | Multi-environment trials | |
|---|---|---|---|---|---|
| | LRT Genotype | LRT Genotype | LRT Genotype | LRT Genotype | LRT GEI |
| WF | 120.79* | 9.52* | 103.09* | 191.77* | 71.02* |
| DF | 95.74* | 11.7* | 89.83* | 132.66* | 82.97* |
| LF | 92.78* | 37.93* | 104.96* | 212.9* | 51.64* |
| TS | 51.62* | 16.99* | 157.5* | 217.88* | 14.18* |
| WS | 113.81* | 19.35* | 126.97* | 218.51* | 73.31* |
| WP | 98.4* | 11.83* | 53.44* | 138.96* | 51.6* |
| YP | 39.37* | 15.38* | 92.53* | 159.84* | 4.04* |
| SS | 24.68* | 28.71* | 75.73* | 153.44* | 29.18* |
| Yield | 26.44* | 5.57* | 71.95* | 97.93* | 49.51* |

WF = weight of fruit; DF = diameter of fruit; LF = length of fruit; TS = thickness of skin; WS = weight of skin; WP = weight of pulp; YP = yield of pulp; SS = total soluble solids.

*Significant according to the $\chi^2$ test ($\alpha$ = 0.05).

for all the environments with no covariance) to the residual effect. For TS and YP, the VCOV matrices that resulted in the lowest AIC and BIC values were CSHet (heteroscedasticity and same covariance among genotypes) for **G**, and DIAG (heterogeneous variances between environments and no covariance) for **R**. For WP, the best VCOV for **G** was Ar1H (approximates UNST, but with a reduced the number of estimated parameters), and DIAG for **R**. Finally, for SS, the selected VCOV matrices were CSHom (homoscedasticity for genetic effects and same covariance among genotypes) for **G**, and DIAG for **R**. Once the most appropriate models were obtained, statistical analysis was performed in order to estimate the genetic parameters. Since fruit ripening was not synchronous for the genotypes, the covariate 'days to harvest (DH)' was also added to the models. However, no significant differences were found (data not shown).

The MET analysis suggested that the environment significantly affected all traits, except WS. This implies that traits vary with the environment or that both locality and crop season influence the performance of fruit traits (Table 2). Furthermore, the random effect of GEI significantly affected all traits. These findings also indicate that genotypes do not show consistent behavior across environments, and this should be taken into account when selecting genotypes (Table 1).

Mean phenotypic values and the range of values, heritability, coefficient of variation and genetic, phenotypic and residual variances for each of the nine traits are summarized in Table 2. The mean values of the full-sibs across environments is higher than that of the parents, SV3 and 2(12), for all traits except SS and TS. The generalized measurement of heritability, proposed by [19], varied considerably from one trait to another and one environment to another, ranging from 0.41 (Yield, environment B) to 0.94 (TS, environment C). Comparing the heritability estimates for different environments, the values for C (season 2015–16) were the highest (except for YP), followed by those of A and B (2014–15 and 2015–16, respectively, both in the same locality). Even though there are exceptions (e.g. Yield), these results show that much of the observed phenotypic variation can be attributed to genetic differences. Regarding the MET analyses, overall high heritabilities were found, especially when compared to the single-environment trial for A and B (Table 2).

Although studying a semi-perennial species using large experimental areas (~2 ha per trial), relatively low CV values were obtained for all traits, ranging from 5.32% (SS, environment C)

**Table 2. Means, Amplitudes and Estimates of Genetic, Genetic by environment interaction, phenotypic and residual variances ($\hat{\sigma}_g^2$, $\hat{\sigma}_{ge}^2$ $\hat{\sigma}_f^2$, $\sigma^2$, respectively), coefficient of variation (CV%) and heritability ($H_c^2$) for nine fruit traits, assessed in a full-sib population of sweet fassion fruit and its parents in three environments (A, B and C).**

| | | | Multi-environment trials | | | | | | |
|---|---|---|---|---|---|---|---|---|---|
| Trait | Mean-parents | Mean-full-sibs | Amplitude-full-sibs | $\hat{\sigma}_g^2$ | $\hat{\sigma}_{ge}^2$ | $\hat{\sigma}_f^2$ | $\hat{\sigma}^2$ | CV (%) | $H_c^2$ |
| WF | 147.20 | 169.44 | (78.44–375.73) | 839.15 | 593.32 | 2393.28 | 960.82 | 18.3 | 0.86 |
| DF | 67.79 | 69.15 | (53.67–92.50) | 8.53 | 11.77 | 37.95 | 17.65 | 6.1 | 0.77 |
| LF | 106.66 | 112.71 | (88.00–149.50) | 33.90 | 22.68 | 107.52 | 50.94 | 6.3 | 0.84 |
| TS | 8.72 | 8.86 | (5.00–14.00) | 0.78 | 0.15 | 1.77 | 0.83 | 10.3 | 0.90 |
| WS | 108.53 | 119.49 | (41.99–311.06) | 636.45 | 408.48 | 1635.85 | 590.93 | 20.3 | 0.88 |
| WP | 38.66 | 49.95 | (2.38–111.25) | 49.47 | 45.44 | 214.36 | 119.45 | 21.9 | 0.78 |
| YP | 25.91 | 29.83 | (1.95–52.19) | 13.12 | 6.82 | 44.90 | 24.96 | 16.7 | 0.83 |
| SS | 16.89 | 16.03 | (10.00–21.70) | 0.83 | 0.10 | 2.01 | 1.08 | 6.5 | 0.89 |
| Yield | 2.12 | 2.36 | (0.10–12.82) | 0.89 | 1.19 | 4.71 | 2.63 | 68.8 | 0.73 |
| | | | Single-environment trials | | | | | | |
| | | | Environment A | | | | | | |
| Trait | Mean-parents | Mean-full-sibs | Amplitude-full-sibs | $\hat{\sigma}_g^2$ | | $\hat{\sigma}_f^2$ | $\hat{\sigma}^2$ | CV (%) | $H_c^2$ |
| WF | 125.18 | 156.41 | (78.44–375.73) | 2635.10 | | 3575.17 | 940.07 | 19.6 | 0.76 |
| DF | 64.56 | 67.74 | (53.67–92.50) | 32.21 | | 52.11 | 19.90 | 6.6 | 0.73 |
| LF | 105.48 | 111.93 | (88.00–149.50) | 79.66 | | 135.92 | 56.20 | 6.7 | 0.72 |
| TS | 7.40 | 8.89 | (5.00–14.00) | 0.64 | | 1.69 | 1.05 | 11.5 | 0.64 |
| WS | 86.11 | 110.34 | (46.22–311.06) | 1736.24 | | 2335.35 | 599.11 | 22.2 | 0.76 |
| WP | 39.08 | 46.08 | (15.84–111.25) | 150.85 | | 250.43 | 99.58 | 21.7 | 0.73 |
| YP | 31.14 | 29.55 | (13.37–47.73) | 7.34 | | 31.08 | 23.74 | 16.5 | 0.54 |
| SS | 14.16 | 15.67 | (10.00–19.20) | 0.56 | | 1.85 | 1.29 | 7.2 | 0.60 |
| Yield | 2.40 | 1.90 | (0.167–11.49) | 1.73 | | 3.59 | 1.86 | 71.7 | 0.68 |
| | | | Environment B | | | | | | |
| Trait | Mean-parents | Mean-full-sibs | Amplitude-full-sibs | $\hat{\sigma}_g^2$ | | $\hat{\sigma}_f^2$ | $\hat{\sigma}^2$ | CV (%) | $H_c^2$ |
| WF | 155.06 | 157.38 | (88.53–260.88) | 247.18 | | 1084.38 | 837.20 | 18.4 | 0.45 |
| DF | 70.00 | 66.95 | (58.00–81.00) | 5.70 | | 22.41 | 16.71 | 6.1 | 0.48 |
| LF | 115.00 | 109.43 | (90.00–142.00) | 42.57 | | 92.72 | 50.14 | 6.5 | 0.63 |
| TS | 7.78 | 8.15 | (6.00–11.00) | 0.45 | | 1.30 | 0.86 | 11.4 | 0.56 |
| WS | 99.49 | 105.10 | (60.09–193.50) | 238.89 | | 722.45 | 483.55 | 20.9 | 0.55 |
| WP | 55.58 | 52.28 | (2.38–84.23) | 46.22 | | 181.92 | 135.70 | 22.3 | 0.48 |
| YP | 35.99 | 33.30 | (1.95–51.42) | 22.31 | | 55.93 | 33.62 | 17.4 | 0.60 |
| SS | 14.11 | 15.81 | (10.00–18.80) | 0.65 | | 1.87 | 1.22 | 7.0 | 0.56 |
| Yield | 2.14 | 1.20 | (0.107–4.99) | 0.10 | | 0.70 | 0.55 | 64.4 | 0.41 |
| | | | Environment C | | | | | | |
| Trait | Mean-parents | Mean-full-sibs | Amplitude-full-sibs | $\hat{\sigma}_g^2$ | | $\hat{\sigma}_f^2$ | $\hat{\sigma}^2$ | CV (%) | $H_c^2$ |
| WF | 157.23 | 194.53 | (66.45–357.29) | 1345.72 | | 2362.71 | 1016.99 | 16.4 | 0.89 |
| DF | 69.81 | 72.77 | (54.33–88.80) | 20.05 | | 35.94 | 15.89 | 5.5 | 0.89 |
| LF | 107.09 | 116.77 | (93.00–144.00) | 52.18 | | 98.42 | 46.23 | 5.8 | 0.88 |
| TS | 8.99 | 9.54 | (5.00–12.67) | 1.40 | | 1.97 | 0.57 | 7.9 | 0.94 |
| WS | 116.29 | 143.04 | (41.99–287.45) | 1067.24 | | 1687.36 | 620.11 | 17.4 | 0.91 |
| WP | 40.94 | 51.49 | (3.25–95.47) | 76.40 | | 211.30 | 134.90 | 22.6 | 0.79 |
| YP | 25.37 | 26.63 | (2.53–52.19) | 20.15 | | 41.61 | 21.46 | 17.4 | 0.86 |
| SS | 17.41 | 16.62 | (13.33–21.70) | 1.03 | | 1.81 | 0.78 | 5.3 | 0.89 |
| Yield | 2.71 | 3.98 | (0.10–12.82) | 3.67 | | 7.96 | 4.30 | 52.1 | 0.84 |

Heritability estimated according Cullis et al. (2006). WF = weight of fruit; DF = diameter of fruit; LF = length of fruit; TS = thickness of skin; WS = weight of skin; WP = weight of pulp; YP = yield of pulp; SS = total soluble solids.

*MET analyses were based on the interaction model.

to 22.56% (WP, environment C). According to [23], CVs are expected to range from 5 to 15% in field experiments. The predominantly low CVs (<10%) obtained indicate good experimental precision. The only exception was Yield, in which due to particular features of the trait and the methodology used for estimation, high CV values were estimated regardless of the environment. Furthermore, the estimated CVs for each trait in A, B, C, or the MET analysis were very similar, denoting that the experimental accuracy was comparable among environments. For instance, CV values for DF were the lowest in all three environments (6.59% in A; 6.11% in B and 5.48% in C) and in the MET analysis (6.07%).

## Correlation between environments and traits

Pairwise genetic correlations among the nine fruit traits and scatter charts are shown in Fig 2. Based on the values obtained, the correlations were grouped into three classes: weak ($|r| \leq 0.45$), moderate ($0.46 \leq |r| < 0.76$) and strong ($|r| \geq 0.76$). Seven positive correlations were classified as weak (LF-TS, LF-WP, TS-WP, WP-YP, YP-SS, Yield-TS, Yield-LF); ten as moderate (WF-LF, WF-WP, DF-LF, DF-TS, LF-WS, DF-WP, Yield-WS, Yield-DF, Yield-WP); and six as strong (WF-DF, WF-WS, WF-Yield, DF-WS, WS-TS, TS-WS). Most of the negative

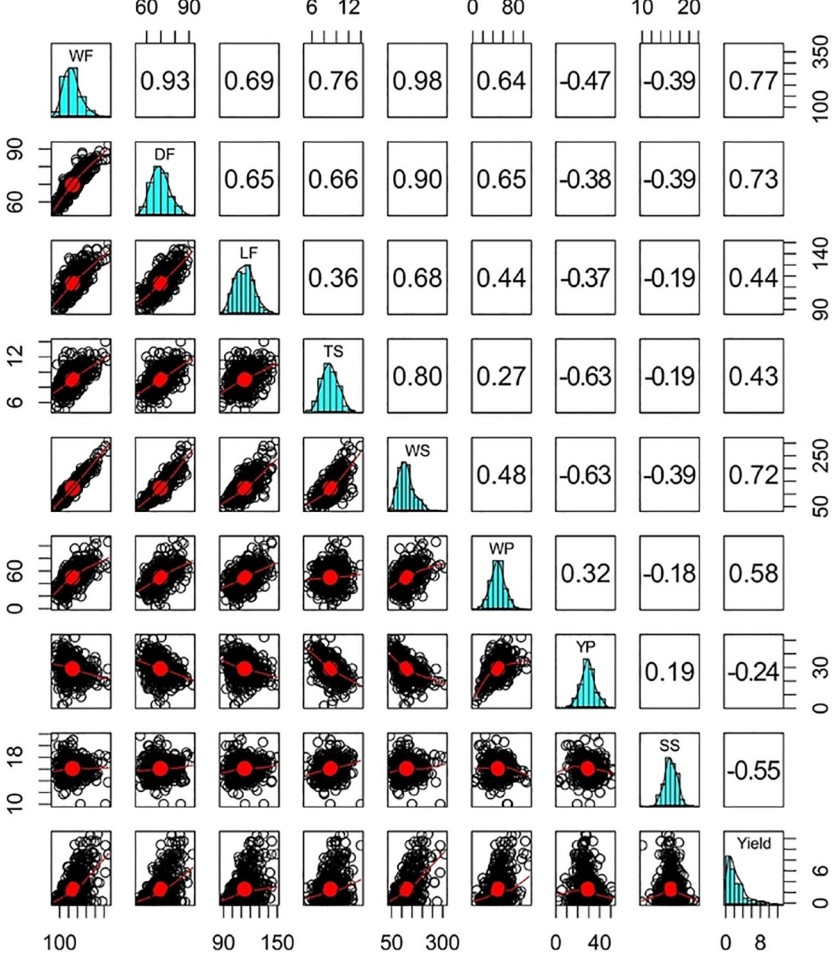

**Fig 2. Histograms (diagonal), scatter charts (below diagonal) and genetic correlations (above diagonal) for fruit traits and yield in a set of full-sibs of sweet passion fruit.** WF = weight of fruit, DF = diameter of fruit; LF = length of fruit; TS = thickness of skin; WS = weight of skin; WP = weight of pulp; YP = yield of pulp; SS = total soluble solids.

correlations were weak, although WF-DF, TS-YP, WS-YP and SS-Yield were moderate. The strongest positive correlation (0.98) was detected between WF and WS, while the strongest negative correlation (0.63) was observed for interactions involving YP and skin traits (YP-WS and YP-TS). Furthermore, although predominantly weak, all interactions involving YP and SS were negative, except YP-WP and YP-SS.

We also investigated the genetic correlation between environments for all traits (S2 Table). Between A and B, $r_{A,B}$ ranged from 0.17 to 0.95 (DF and YP, respectively), whereas between A and C, $r_{A,C}$ ranged from 0.39 to 0.96 (WP and SS, respectively) implying a higher correlation between A and C than between these two environments and B. This pattern of correlation values has implications for genotype ranking, depending on the trait and environment, lending further weight to the existence of GEI.

Also, with the aim of studying genetic correlations and assessing the behaviour of groups of traits, a correlation network was built for each environment. In this analysis, circles represent the traits, line colour indicates positive (green) and negative (red) correlations, and line thickness denotes magnitude (Fig 3). Overall, correlation network plots corroborate the average genetic correlations (Fig 2). Comparing environments, in A the genetic correlations among traits were overall positive and higher (Fig 3a) than those found in B (thinner lines showing weak and moderate correlations–Fig 3b). In C, there was a pattern more similar to that found in A, though not as strong (Fig 3c). Moreover, YP and SS showed weak to moderate negative correlation with most of the other traits for the three environments, especially A. Finally, all traits other than YP and SS showed mainly positive correlations with each other (Fig 3).

## Genotype by environment interaction analysis

GGE biplot analysis was performed in order to provide a comprehensible view of the GEI and allow better interpretation of MET results. This approach is useful for evaluating genotypic performance across environments, comparing different test environments and elucidating how traits are interrelated [21]. The GGE biplot is constructed by plotting the first two principal components (PC1 and PC2) derived from singular value decomposition of the environment-centered data. Briefly, when environments are allocated to different sectors, it means that genotype performance diverges, indicating a crossover GEI pattern. Otherwise, if all the environments are allocated to the same sector, GEI is weaker. In terms of genotype performance, the best genotypes are those located at the polygon vertices.

The first two principal components accounted for 76.56% of the variation (PC1 = 59.78% and PC2 = 16.74%), showing the efficiency of this kind of analysis in explaining most of the

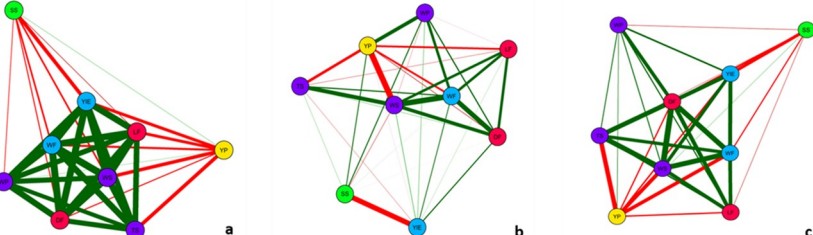

**Fig 3. Correlation network for nine fruit traits in three environments: A, 2015 (a), B, 2016 (b) and C, 2016 (c).**
WF = weight of fruit; DF = diameter of fruit; LF = length of fruit; TS = thickness of skin; WS = weight of skin; WP = weight of pulp; YP = yield of pulp; SS = total soluble solids; YIE = yield. Circles represent traits, and lines represent Pearson correlation coefficients. Green and red lines represent positive and negative correlations, and line thickness indicates the magnitude of the correlation.

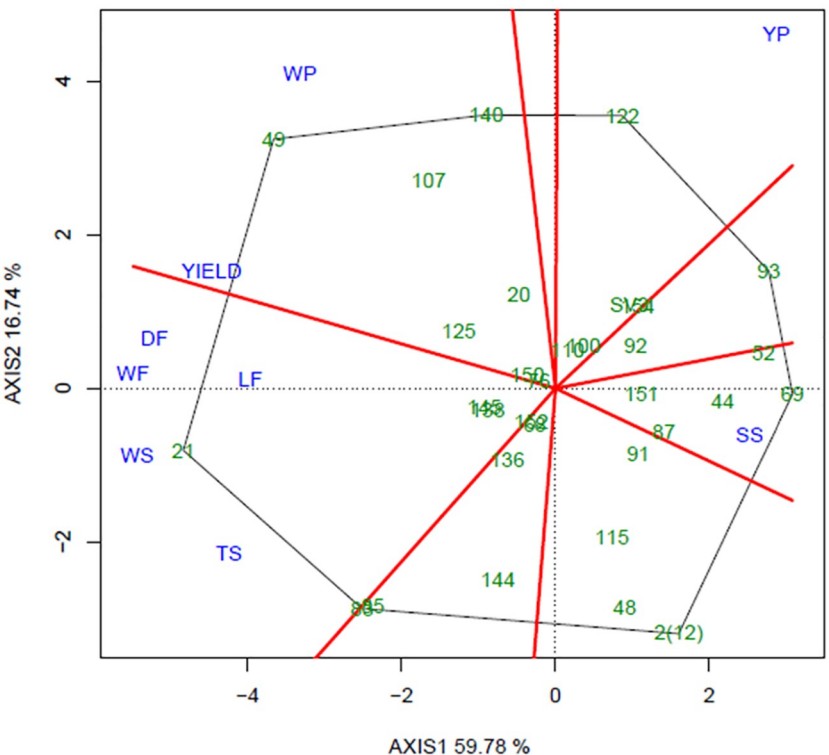

**Fig 4. Polygon view of GGE biplot showing the behavior of 30 genotypes (full-sibs) in respect of nine fruit traits.**
WF = weight of fruit; DF = diameter of fruit; LF = length of fruit; TS = thickness of skin; WS = weight of skin;
WP = weight of pulp; YP = yield of pulp; SS = total soluble solids. Gray lines represent the polygon formed by
genotypes and red lines represent the sectors shared by traits.

variance resulting from all trait data sets (Fig 4). Genotype distribution over the entire graph
and positions in different sectors indicate the existence of high levels of variability within the
population.

The most strongly correlated traits are those related to fruit size and shape (WF, LF, DF,
WS and TS) that are located within a single sector of the polygon (Fig 4). In this sector, geno-
type 21 is positioned at the polygon vertex indicating high performance for these traits. Other
interesting genotypes are 49, 107 and 140, which appear in the sector formed by WP and
Yield; genotype 122 also showed higher YP. On the other side of the biplot, SS was negatively
correlated with all traits, except YP (Fig 4). The SS sector groups genotypes with high SS values,
such as 69, 52 and 44. However, despite the sweetness of the fruit, these genotypes are inferior
in terms of WF, DF, LF, WS, TS, WP and Yield. Regarding the parent plants, while the male
(SV3) is placed near the intersection, and thus of average performance only, the female 2(12) is
positioned at a single polygon vertex (Fig 4). Nevertheless, even though it does not share the
sector with any trait, the positioning of 2(12) in relation to WP and Yield reflects its wild,
unimproved attributes.

In terms of fruit yield in all environments (A, B and C), the best performing genotypes were
21, 49, 136, 52 and 69. For yield specifically, a different genotype performed better in each
environment: 21 in A, 151 in B and 49 in C, showing how influential GEI can be. Additionally,
environment C was more discriminating of genotypes, while environment B did not allow any
clear conclusions to be drawn.

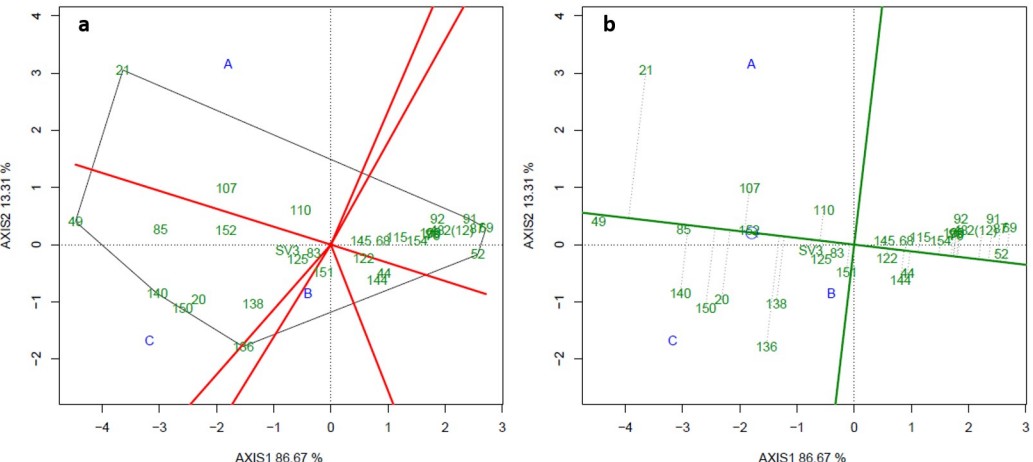

**Fig 5. Polygon view of the GGE biplot showing the behavior of 30 genotypes (full-sibs) in terms of Yield (a).** Average Environment Coordination view of the GGE biplot in three environments: A, 2015, B, 2016 and C, 2016 (b).

The Average Environment Coordination (AEC) view of the GGE biplot is shown in Fig 5b. In the graph, genotype average values in the different environments create an "average environment" point represented by the small blue circle. Genotypes exhibiting high stability are located near this circle, including genotype 152. The projections of the genotype on the abscissa represent the main genetic effects and therefore rank the genotypes in relation to their mean performance. Thus, in accordance with this ranking, the genotypes were classified according to yield, as follows: 49 > 21 > 85 > 140 > 150 > . . . > 151 > overall mean > 145 > 122 > . . . > 52 > 69. The AEC ordinate approximates the contribution of the genotype to GEI, a measurement of stability. Since genotype 152 is located almost on the AEC abscissa with near-zero projection onto the AEC ordinate, it is the most stable genotype. In contrast, 21 and 136 were two of the least stable genotypes. Finally, in terms of the ideal genotype, 49 was ranked at the top and exhibited stable performance (Fig 5b).

The new GGE biplot shown in Fig 6 was based only on the traits used to create the MSI (WP, WS and TS). The reduced size of the polygon compared to those in Figs 4 and 5 reflects lower variability since only three traits are used. WP, WS and TS were grouped into two sectors, one formed by WP in the three environments and other by WS and TS. The polygon vertices are formed by genotypes 49, 21, 85, 2(12), 93 and 122. Once again, genotype 49 was the front-runner with the higher WP values, while 21 was best performer in terms of WS and TS.

## Selection strategy

In fruit crops, the main objective of breeding programs is to increase genetic gains to ensure fruit quality and high yield. In this study, we selected six genotypes from a population of 30 preselected full-sibs [6], representing a selection intensity (SI) of 20%. Table 3 shows response to selection (RS) values that, for mixed models, correspond to BLUP values, and Percentage RS (RS%) is based on four different selection scenarios. The first two scenarios relate to the result obtained if genotypes were selected with the aim of decreasing TS and WS; the third selection is aimed at increasing WP and the forth is based on a MSI for simultaneously selecting for WP and against TS and WS.

In the first scenario (selecting against TS), the six selected genotypes were 93, 69, 44, 52, 140 and 154, and the reduction in TS was 13.2%. Since there is a high correlation between TS and

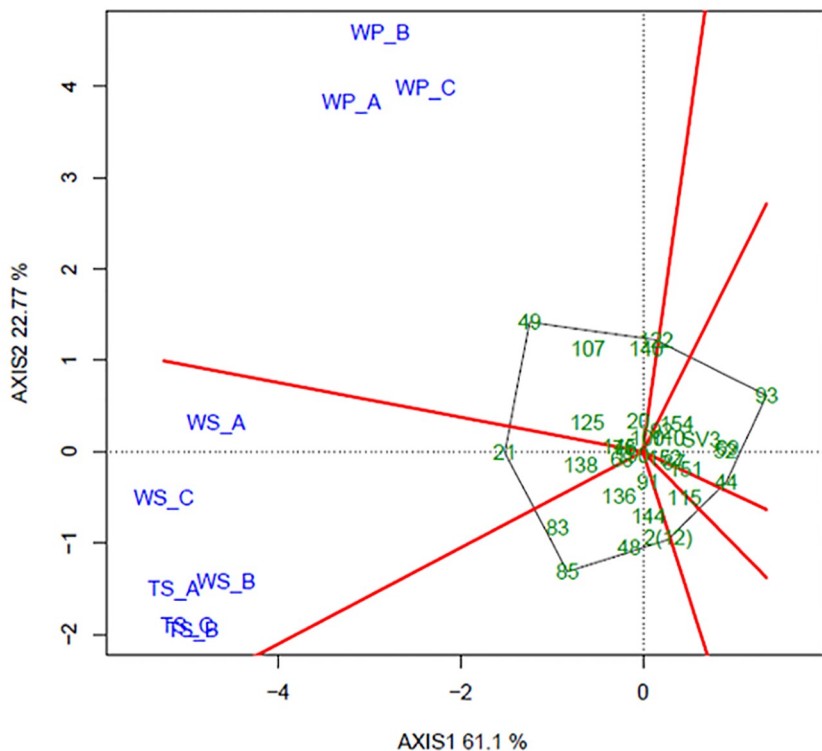

**Fig 6. Polygon view of GGE biplot showing the behavior of 30 genotypes (full-sibs) for traits used to compute the multiplicative selection index.** TS = thickness of skin; WS = weight of skin; WP = weight of pulp in each of environments A, B and C. Gray lines represent the polygon formed by genotypes, and red lines represent the sectors shared by traits.

other traits, there was a decrease in WF, WS and Yield (18.6%, 23.6% and 13.3%, respectively), as well as a significant increase in YP (11.4%). The second scenario (selecting against WS) produced similar results. The selected genotypes were 69, 93, 52, 44, 87 and 122, and the decrease in WS was 26.1%; other traits were also reduced, including WF and WS, and especially Yield

**Table 3. Response to Selection (RS) for nine fruit traits in four selection scenarios: Selection was performed for individual traits separately, in order to reduce the thickness and weight of fruit skin (TS and WS) and increase the weight of pulp (WP), or based on a multiplicative selection index (MSI).**

| Trait | Selection against TS | | Selection against WS | | Selection for WP | | MSI | |
|---|---|---|---|---|---|---|---|---|
| | RS | RS% | RS | RS% | RS | RS% | RS | RS% |
| WF | −32.033 | −18.580 | −37.559 | −21.786 | 38.643 | 22.414 | 38.643 | 22.414 |
| DF | −3.322 | −4.769 | −4.537 | −6.513 | 4.223 | 6.063 | 4.223 | 6.063 |
| LF | −3.739 | −3.296 | −4.571 | −4.030 | 3.968 | 3.498 | 3.968 | 3.498 |
| WS | −29.038 | −23.645 | −32.122 | −26.156 | 25.864 | 21.061 | 25.864 | 21.061 |
| TS | −1.188 | −13.184 | −1.129 | −12.532 | 0.395 | 4.389 | 0.395 | 4.389 |
| WP | −3.048 | −6.146 | −5.171 | −10.426 | 12.385 | 24.972 | 12.385 | 24.972 |
| YP | 3.327 | 11.429 | 2.619 | 8.997 | 1.681 | 5.773 | 1.681 | 5.773 |
| SS | 0.129 | 0.804 | 0.312 | 1.941 | −0.242 | −1.503 | −0.242 | −1.503 |
| Yield | −0.347 | −13.346 | −0.793 | −30.448 | 1.081 | 41.510 | 1.081 | 41.510 |

WF = weight of fruit; DF = diameter of fruit; LF = length of fruit; TS = thickness of skin; WS = weight of skin; WP = weight of pulp; YP = yield of pulp; SS = total soluble solids.

**Table 4. Results of self-pollinations and all reciprocal crosses involving six selected genotypes of sweet passion fruit.**

| Female parent | Male parent | | | | | |
|---|---|---|---|---|---|---|
| | 21 | 49 | 107 | 122 | 125 | 140 |
| 21 | – | + | – | + | + | + |
| 49 | + | – | + | 40% | + | + |
| 107 | 26.7% | + | – | + | + | + |
| 122 | + | + | + | – | 25% | 11.1% |
| 125 | + | + | 25% | 25% | – | – |
| 140 | + | + | + | – | 7.7% | – |

(+) Compatible cross resulting in over 50% fruit set; (–) Incompatible cross with no fruit set; (%) Percentage in which fruit set was observed but below 50%.

(21.8%, 26.1% and 30.4%, respectively); YP was increased (9%). The third scenario selected for WP, and 49, 107, 122, 140, 21 and 125 were the selected genotypes. The response to selection was positive for all traits, except SS. In addition to WP (24.9%), the highest gains were obtained for WF, WS and Yield (22.4%, 21% and 41.5%, respectively). Finally, in the fourth scenario, using the MSI resulted in the same set of genotypes as in the third scenario. However, the ranking was different (49, 21, 107, 125, 140 and 122). As a consequence of selecting the same genotypes, all responses to selection were the same as in the third scenario.

## Reproductive compatibility between selected genotypes

Based on the MSI, six genotypes were selected: 21, 49, 107, 122, 125 and 140 (S3 Table, S1 Fig). Since these genotypes are full-sibs, it is essential to test whether they can be crossed with each other and set fruits. Furthermore, although *P. alata* is assumed to be self-incompatible [24] as corroborated by molecular marker-based studies [4], the crossing studies necessary to prove this hypothesis have not been conducted. For this reason, we carried out a complete 6 × 6 diallel crossing, with reciprocals and self-pollinations, involving all the selected genotypes.

Our results show that most of the crosses were compatible (over 50% fruit set). This value is the percentage of pollinated flowers ultimately forming fruits. As expected, the occurrence of self-incompatibility within the species was evidenced by the absence of fruit set in all self-pollinations. Some of the crosses also did not produce fruits (e.g. 21 × 107, 125 × 140 and 140 × 122), possibly because of genotype relationships. In addition, there were cases in which fruits were produced but at rates lower than 50% (49 × 122, 107 × 21, 122 × 125, 122 × 140, 125 × 107, 125 × 122, and 140 × 125), indicating partial compatibility (Table 4). It is worth noting that all genotypes produce fruits if used as females, which is essential to guarantee yields in commercial orchards.

## Discussion

Despite the great potential and the prospects for higher consumption and utilization of *P. alata* as a fruit crop, production remains low and unstable, mainly due to the lack of improved varieties. In passion fruit, like other fruit crops, the main purpose of breeding is to meet the demand for quality [6, 25, 26]. According to [27], sweet passion fruits should weigh 200 to 300 g, be oval in shape, free from apical softening, with a firm skin, rich pulp yield (over 30%), high sugar content and significant yield. However, all these goals are not easily achieved in a few selection cycles, requiring a medium-term program that takes into account correlations between these traits.

To face this challenge, our research group conducted breeding programs using modern quantitative genetics to generate information and select genotypes that can lead to the development of varieties with improved fruit quality and yield. To do this, a segregating population ($F_1$) was developed by crossing two outbred, divergent accessions of *P. alata*. In parallel, this full-sib family (n = 180) was genotyped using molecular markers [28] and used to construct a unique linkage map for the species [29]. One hundred individuals were then sampled and field-evaluated, and QTL mapping analysis performed to identify loci associated with fruit quality traits. The MSI was also used to select the 30 most promising genotypes [6].

In the present study, we reevaluated these 30 full-sibs in three environments and estimated genetic and phenotypic parameters for nine fruit traits to determine if there was still some genetic variability within the selected population for continuing the breeding process. The six most promising genotypes were then selected and their fruit set capability evaluated.

To summarize, linear mixed models were applied to analyze the MET so that several VCOVs could be investigated in terms of the **G** and **R** matrices and each trait. Based on the AIC and BIC values, the best models for WF, LF and WS were UNST (unstructured) for **G** and ID for **R**. For TS and YP, CSHet was the best for **G**, and DIAG for **R**. For WP, Ar1H was used for **G** and DIAG for **R**. Finally, for SS, CSHom was used for **G**, and DIAG for **R** (Table 1).

As is usually the case for datasets with complex GEI, our results show that none of the simplest VCOV, ID and DIAG matrices were selected for **G**. Furthermore, breeding data are frequently unbalanced since diversified sets of genotypes are evaluated in different trials [30,31]. Statistical methods should therefore be used to model different variances and covariances between environments. Thus, approaches that model complex VCOV matrices, such as UNST, are better because they can capture both the heterogeneity of genetic variance and complex covariance structures, resulting in a more accurate prediction of single- or multi-environment trials. However, it is important to note that the number of estimated parameters in unstructured models can inflate rapidly as the number of trials increases, which can make UNST models less parsimonious, requiring alternative VCOV models when analyzing moderate to large MET datasets [32,33].

The low CV values observed herein for all traits show the high precision of the environmental conditions, and these values are particularly interesting due to the semi-perennial behavior of passion fruit orchards and the large experimental areas used (over 2 ha per trial). The exceptions were the high Yield CVs, which may be trait-intrinsic but could also be attributable to the method used to estimate Yield, which was based on both the number of fruits per plant and the average weight of the fruits. In addition, the low mean values, especially those obtained for environments A and B, lead to higher CV values, since this measurement represents the ratio of the standard deviation to the mean, the denominator of the CV equation (Table 2).

For most traits the mean values of the full-sibs across environments exceeds the mean values of the parents. It shows that, since the 30 full-sibs we evaluated in this study are result of a previous selection, transgressive genotypes have been selected [6]. High heritabilities ($H^2 \geq$ 0.50) were estimated for each environment and for the MET analyses, reaching values up to 0.94 (SS) (Table 2). Although heritabilities were highest overall in environment C, low values were found in B, especially for Yield (0.41). In the previous population (n = 100), high heritabilities for the same fruit traits (except Yield) were estimated, varying from 0.59 (WP) to 0.82 (SS) [6]. These significant broad sense heritability values are particularly important since the species can be propagated by cuttings and thus all types of genetic variance can be exploited to predict responses to selection [34].

Genetic correlations between traits showed mostly intermediate to high values, especially for correlations associated with fruit size and shape. The highest values (some exceeding 0.76)

were found among WF, DF, WS, TS and Yield (Fig 2). These findings are also supported by the correlation networks, especially in environments A and C (Fig 3).

According to [35], DF and LF are strongly correlated with each other and with TS, but no correlation with YP was found, indicating that larger fruits in *P. alata* populations do not necessarily have higher pulp content. For *P. edulis*, there are several reports of negative correlations between TS and WF, WP, DF, LF, number of fruits and Yield [26,36–38] enabling breeders to successfully select against TS.

Comparing the genetic correlations found herein (n = 30) with those reported for the population studied by [6] (n = 100), there were some differences in correlation magnitudes. Analyzing in detail the traits that comprise the MSI (TS, WS and WP), we found significant variation in the correlation between WP and WS; the initial value was 0.55 but dropped to 0.48 after selection. In addition, the correlation between WP and TS dropped from 0.28 to 0.27. This occurred because the selection was for WP and against both WS and TS. What is particularly interesting is that these reductions in the magnitudes of correlations were obtained with only one cycle. If there are strong correlations, selection based on a single trait might result in an increment of undesirable traits, as occurred for WS and TS when individuals were selected for WP. For breeders, even a slight detachment of correlated traits is of great interest, since it allows selection for one trait with little impact on the others, denoting that using a selection index might be appropriate when highly correlated traits are targeted in breeding programs.

For all the traits we evaluated (except WS), the MET analysis indicated that the effect of the environment was significant. There was also a significant random effect of GEI, indicating that genotypes do not perform consistently across environments. The GEI was then analyzed and interpreted by GGE biplot. Our findings revealed the existence, albeit small, of variability in the population, corroborating [6], especially for WF and WS. Furthermore, the strong correlation among traits was also confirmed by this method. In addition, some of the genotypes subsequently selected by the MSI were significant, with positions at polygon vertices (21, 49, 122 and 140) (see Fig 4).

Yield performance across environments was also revealed by the GGE biplot model, showing the importance of this trait in the GEI. The analysis indicated that genotypes 21, 49, 52, 69 and 136 were the most promising (Fig 5a). The AEC view of the GGE biplot allowed us to study the stability of genotypes across environments (Fig 5b) and showed that, for Yield, the most stable genotype was 152, while 21 and 136 were two of the less stable genotypes.[39] proposed that the ideal genotype must have both high average performance and high stability within a mega-environment. In our analysis, in contrast to 21 and 136, which were unstable despite the high yield, genotype 49 showed high yield and a very stable pattern. It is therefore a promising candidate for selection. Still on the subject of Yield, in all three environments low estimates were obtained, averaging 1.9 (A), 1.2 (B), 3.98 (C) and 2.98 (MET analysis) tonnes per ha (Table 2). Although these values are relatively low, it is worth noting that they represent average phenotypic values for the entire population, since in terms of Yield, the most promising genotypes (21, 49 and 136) produced maximal values of 11.5 (A), 5.0 (B) and 12.8 (C) tonnes per ha.

In an attempt to select superior individuals, we applied a selection intensity of 20% and simulated four scenarios seeking to decrease skin-related traits and increase WP. Because of the high correlations between traits, selection based on the MSI in environment (C) produced the most satisfactory results, optimizing WP gain and TS and WS losses. For example, if selection was applied only against TS, the thickness of fruit skin would decrease 13.2% (Table 3), but other important traits such as Yield would also be significantly impaired (13.3%). Comparing the results with those obtained in the source population [6], higher percentages of selection gain for all traits were achieved by using the MSI. Furthermore, selection based on the MSI

was the only method that resulted in higher WP and lower TS and WS. Thus, according to the MSI, the selected genotypes were: 49, 21, 107, 125, 140 and 122 (S3 Table, S1 Fig).

GGE biplot analysis was also performed using only MSI traits. Again genotype 49 was the best genotype, with high WP. Moreover, 21 and 122, selected by MSI, were also positioned at the polygon vertices, as were 107 and 140. Although some genotypes, such as 49 and 122, were highly stable for yield (Fig 5b), when compared on a performance basis, WS, TS and WP in the three environments were ranked differently, reflecting the complex GEI interaction (S2 Fig).

Pulp yield determines how much of the weight of the fruit can be attributed to the weight of the pulp. As mentioned above, *P. alata* is almost wild and its low YP is the result of its heavy skin and low pulp content. For the previous population (n = 100), the estimated and expected YP values were 22.43% and 23.37% [6]. However, in our study, YP values were even higher using the MSI, reaching 29.6% (A), 33.3% (B) and 26.6% (C). These YP gains are high, and similar results close to the 30% proposed as ideal for the species have been obtained in other breeding populations [27,35,40].

Since the selected genotypes (49, 21, 107, 125, 140 and 122; S3 Table, S1 Fig) belong to a full-sib family, diallel crosses were carried out in all possible reciprocal plant combinations to check their ability to produce fruits. Cross-compatibility of the selected genotypes is essential to continue with breeding programs, and even provide genotypes to farmers with commercial orchards.

We have provided evidence of self-incompatibility in *P. alata*, confirming previous findings obtained using molecular markers [4]. For all reciprocal combinations, 10% (3/30) were found to be incompatible and 23% (7/30) partially compatible. Importantly, most of the combinations were found to be compatible (20/30) and all the six genotypes produced fruits if used as females.

We also noticed differences in reciprocal crosses, corroborating other studies on yellow passion fruit [3,41,42]. In our study, all the reciprocals of the incompatible crosses (♀21 × ♂107, ♀125 × ♂140 and ♀140 × ♂122) had low rates of fruit production (♀107 × ♂21, ♀140 × ♂125 and ♀122 ×♂140). These results lend weight to the idea that there is genetic control of self-incompatibility in *Passiflora*, which has already been described as homomorphic-sporophytic [43] and gametophytic-sporophytic [3].

According to [43], the incompatibility mechanisms in *Passiflora* represent a direct challenge to breeders if they are to produce hybrids, release synthetic varieties and establish clones. The genotypes used to set up commercial orchards must be very carefully chosen in order to guarantee highly efficient pollination. In this study, the predominance of compatible crosses indicates that genotypes 21, 49, 107, 122, 125 and 140 could be used, for instance, to produce a recurrent selection population for increasing the frequency of favorable alleles involved in genetically controlling fruit quality traits and yield, and even recommended to farmers.

In conclusion, this study shows that many of the phenotypic differences are due to genetic variation, allowing high heritability estimates. Although strong genetic correlations were detected for most traits, we were able to demonstrate that the use of a selection index could help reduce the magnitudes of correlations between desirable and undesirable traits. This index allowed the selection of six promising genotypes that are also mostly cross compatible and therefore can be used commercially or for continuing the breeding program. Finally, our results provide a comprehensible view of the genotype by environment interaction and allowed us to interpret how the sweet passion fruit genotype performs across environments.

## Supporting information

**S1 Table. Akaike (AIC) and Bayesian (BIC) Information Criteria and number of parameters (nPAR) of the multi-environment models tested in respect of different genetic (G)**

and residual (R) variance-covariance structures for nine fruit traits of 30 genotypes (full-sibs) of sweet passion fruit. WF: weight of fruit; FD: diameter of fruit; LF: length of fruit; TS: thickness of fruit skin; WS: weight of fruit skin; WP: weight of pulp; PY: pulp yield; SS: total soluble solids; ID: identity; DIAG: diagonal; CShom: homogeneous compound symmetry; Ar1: first order autoregressive; Ar1H: heterogeneous first order autoregressive; CSHet: heterogeneous compound symmetry; UNST: unstructured; FA1: first order factor analysis.
(XLSX)

**S2 Table. Genetic correlations between the locations A, B and C and the respective deviations (DS) for eight traits related to fruit quality and yield.** WF: weight of fruit; FD: diameter of fruit; LF: length of fruit; TS: thickness of fruit skin; WS: weight of fruit skin; WP: weight of pulp; PY: pulp yield; SS: total soluble solids.
(XLSX)

**S3 Table. Ranking of 32 genotypes (with parents included) after applying the Multiplicative Index (MI) with the aim of increasing WP and decreasing TS and WS.**
(XLSX)

**S1 Fig. Response to selection in sweet passion fruit.** Note the difference in the thickness of skin and pulp content between fruits of the parental accessions (above). The six superior genotypes (below) were selected on the basis of a multiplicative index (MI). Bar = 1 cm.
(PDF)

**S2 Fig. Predicted genetic values of the six selected genotypes for Weight of Skin (a), Thickness of Skin (b) and Weight of Pulp (c) in each of environments A, B and C.**
(PDF)

## Acknowledgments

We thank Mr. Steve Simmons for proofreading the manuscript and Dr. João Paulo Rodrigues Marques for helping us with the stereomicroscope and digital imaging.

## Author Contributions

**Conceptualization:** Maria Lucia Carneiro Vieira.

**Formal analysis:** Lourdes Maria Chavarría-Perez, Kaio Olimpio Graças Dias, Guilherme Silva Pereira, João Ricardo Bachega Feijó Rosa.

**Funding acquisition:** Maria Lucia Carneiro Vieira.

**Investigation:** Lourdes Maria Chavarría-Perez, Willian Giordani, Zirlane Portugal Costa, Carolina Albuquerque Massena Ribeiro, Anderson Roberto Benedetti, Luiz Augusto Cauz-Santos.

**Methodology:** Lourdes Maria Chavarría-Perez, Kaio Olimpio Graças Dias, Guilherme Silva Pereira.

**Supervision:** João Ricardo Bachega Feijó Rosa, Antonio Augusto Franco Garcia, Maria Lucia Carneiro Vieira.

**Writing – original draft:** Lourdes Maria Chavarría-Perez, Willian Giordani, Zirlane Portugal Costa.

**Writing – review & editing:** Maria Lucia Carneiro Vieira.

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
