## [Decision Letter · Decision Letter 0]

10 Oct 2019

PONE-D-19-24935

Improving yield and fruit quality traits in sweet passion fruit: evidence for genotype by environment interaction and cross-compatibility in selected genotypes.

PLOS ONE

Dear Professor Vieira,

Thank you for submitting your manuscript to PLOS ONE. After careful consideration, we feel that it has merit but does not fully meet PLOS ONE’s publication criteria as it currently stands. Therefore, we invite you to submit a revised version of the manuscript that addresses the points raised during the review process.

Some work to do here! Bothe reviewers will be reinvited

We would appreciate receiving your revised manuscript by Nov 24 2019 11:59PM. To enhance the reproducibility of your results, we recommend that if applicable you deposit your laboratory protocols in protocols.io, where a protocol can be assigned its own identifier (DOI) such that it can be cited independently in the future. For instructions see: http://journals.plos.org/plosone/s/submission-guidelines#loc-laboratory-protocols

We look forward to receiving your revised manuscript.

Kind regards,

David A Lightfoot, PhD

Academic Editor

PLOS ONE

**Journal Requirements:**

4.We note that you have not included  financial disclosure in your online submission form;

a)    Please provide a Funding Statement that declares *all* the funding or sources of support received during this specific study (whether external or internal to your organization) as detailed online in our guide for authors at http://journals.plos.org/plosone/s/submit-now.  

b)    Please state what role the funders took in the study.  If any authors received a salary from any of your funders, please state which authors and which funder. If the funders had no role, please state: "The funders had no role in study design, data collection and analysis, decision to publish, or preparation of the manuscript."

c)   If the study was unfunded, please provide a statement that clearly indicates this, for example: "The author(s) received no specific funding for this work.

5. We note that you have not included a competing interests statement:

Please complete the competing interests section fully.  If NO authors have competing interests, please enter: "The authors have declared that no competing interests exist."

If Authors have competing interests please enter competing interest details beginning with this statement:

"I have read the journal's policy and the authors of this manuscript have the following competing interests: [insert competing interests here]"

**Comments to the Author**

1. Is the manuscript technically sound, and do the data support the conclusions?

Reviewer #1: Partly

Reviewer #2: Partly

2. Has the statistical analysis been performed appropriately and rigorously? 

Reviewer #1: Yes

Reviewer #2: Yes

3. Have the authors made all data underlying the findings in their manuscript fully available?

Reviewer #1: Yes

Reviewer #2: Yes

4. Is the manuscript presented in an intelligible fashion and written in standard English?

Reviewer #1: No

Reviewer #2: Yes

5. Review Comments to the Author

Reviewer #1: General comments

The authors have reported on “Improving yield and fruit quality traits in sweet passion fruit: evidence for genotype by environment interaction and cross-compatibility in selected genotypes.” The experimental design is generally adequate, and the flow of ideas is satisfactory. However, there is need a to succinctly define objective(s) of the study. Also, the manuscript seems to report on two separate studies which do not seem to converge into a coherent story. This situation masks the effectiveness of either. While the test for ‘genotype by environment interactions’ is fairly well developed, results reported and discussed, the less well developed ‘compatibility test’ takes up much of the conclusion.

The compatibility test results in some successful fruit formation. However, for the compatibility experiment to have been useful as part of this manuscript, the resulting progeny from these crosses could have been tested side by side with the other 30 ‘individuals’ in the different environments. As it is, the compatibility test is obviously a study of interest to breeders but is unfortunately out of place in this manuscript. I would suggest that it be developed as an independent study.

In all the section, a glaring deficiency is the lack of clarity and specificity of statements made; some of these require adequate citations. Besides statistical analyses and formulae terms should be adopted to the existing study experimental design.

The discussion section needs to be developed beyond simply reiterating what is reported in the results section. I am sure the quality of this manuscript will significantly improve if the following concerns are addressed and grammatical mistakes taken care of.

Abstract

Line 31: popularly not popular

Line 32: “... is assumed to be a self-incompatible species ….” This portion of the sentence is unnecessary unless the ‘self-incompatibility’ is defined. I would leave it out. I believe it is still ok to include the sentence portion “…is highly appreciated for its typical aroma and flavor characteristics” without the first portion.

Lines 33 -34: “With the aim of estimating the genetic and phenotypic parameters related to fruit traits…” Please be clear what genetic and phenotypic ‘parameters’ one should expect in the manuscript.

Lines 34 – 35: Please define the objective of this study succinctly. Hint: you can summarize what is in lines 104 – 107 (under ‘Introduction’).

Lines 36 - 37: ‘In this study, we reevaluated these superior genotypes in three environmental conditions. ----Q: Superiority based on what? Are these the 100 or the 30 genotypes? This is confusing if considered with: Lines 37 – 39: “The results of the multi-environment trial analysis indicate that the genotypes do not behave consistently across these environments, and this was taken into account when selecting genotypes.” …..Q: Were the trials conducted for the 100 or the 30 genotypes before ‘xy’ number of genotypes were selected? To remedy this confusion, I suggest lines 36 – 39 be rewritten…..*

Line 39 – 41: “Pairwise genetic correlations among the fruit traits were evaluated, and different genotype rankings obtained depending on the trait and environment, providing further evidence of genotype by environment interaction.” Up to this point, I do not know ‘the traits’ you are referring to. Please mention the traits in line 34. Also, ranking based on…. what? Mention that basis here, instead of: ‘...depending on the trait and environment’. Please remove: ‘...providing further evidence of genotype by environment interaction’; there is no statement to support this in the abstract.

Lines 41 – 42: “Finally, we used a multiplicative selection index to select 20% of genotypes…….”. Is this 20% of 30 genotypes mentioned in Line 35? Would that mean six (6) genotypes mentioned in line 217? Please be clear.

Lines 44 – 47: “The consensus is that open-pollinated populations can be used as commercial varieties in crop species that are sensitive to inbreeding depression or within breeding programs that are not well developed. For these reasons...” ---------what purpose does this statement serve considering the study objective (which you have not defined succinctly)? Please remove it; otherwise it belongs in other sections e.g., introduction or discussion sections.

Introduction

This section is generally well developed. Except:

Lines 107 to 112: I suggest you remove these sentences. They are unnecessary here.

*Line 109 – 112: Was this one of the objectives of the study, or was it just part of a necessary procedure to enable the evaluation of the actual objective? If so, I suggest you restrict it to the materials and methods. Otherwise, remove it.

Materials and Method

General: Please number the equations so it is easy to reference them when needed.

Please start by rewriting the statement in lines 124 – 127, thus: In this study, we examined 32 genotypes, consisting of a sample (n = 30) of full-sib progeny of sweet passion fruit and the two parents. The two parents are outbred and divergent accessions. Then continue from line 116: “The male parent……………”

Line 115: “Instead of ‘population (N= 30)’, use: sample (n = 30)

Line 124: Describe the progeny, e.g., The 30 full-sibs were part of a progeny of 100 individuals that had been evaluated previously in two ….continue with line 127. Also see comments elsewhere in the discussions.

Line 129: replace “three environments” with: two locations and two seasons, consolidated for a total of three environments (A, B and C) for the purpose of this study. Environment A and B were represented by seasons …………… While Environment C was represented by 2nd season (dates) at location…..…….

Lines 130 – 133: Please name the locations in addition to the grid references.

Lines 143 – 146: Please include the manufacture’s name in addition to instrument specification.

Lines 157 – 167: In the first two model equations, a ‘blocking’ variable has been used, while in the experimental design, it is not explicit what the blocks are. Advice: please adapt the equations to the experimental design; what you have here looks a lot more generic. Either the blocks will need to be defined considering the existing experimental design, or you need to modify the equations to suit your design. This is important.

Line 180 – 181: Briefly explain the importance (justification for) of using AIC and BIC for VCOV structure variables in your LMM analysis. A single sentence will do.

Line 196: Please use small ‘s’ not ‘sigma’ in the equation for CV for your sample distribution.

Line 197: “…the phenotypic average for the sample” instead of “the average for the population”. Remember you are working with sample means. Fortunately, in this particular case, the CV values are not going to change.

Line 215: “…incompatibility”. Please define this term. Did you mean reproductive incompatibility?

Line 216 - 218: “As potential incompatible crosses might occur due to the existence of incompatibility mechanisms in P. alata, the selected genotypes were crossed and fruit set evaluated at the same site (C), using a complete 6 × 6 diallel cross during the 2017/18 and 2018/19 growing seasons” ------Please consider rewriting this sentence. Example: Due to the potential for reproductive self-incompatibility in P. alata, the selected genotypes were crossed in a 6 × 6 diallel design, and fruit set evaluated at the same site (C), during the 2017/18 and 2018/19 growing seasons.

Line 219: Remove ‘artificially’.

Line 225 – 226: What are “10 pollinations”? Instead, write how many (female) plants were pollinated.

Line 228 – 229: Were there some crosses producing 0 < fruit set < 50? How were these classified, or were they simply ignored (assume the reader has not looked at results - Table 4)

Results

Lines 236 – 250: See comments for lines 180 – 181.

Line 252 with reference to Table 2: what is the difference between WP and YP (how do you determine WP as opposed to YP)? Is the DF longitudinal or lateral? Please add this information in the materials and method under ‘….Measurements’ (Line 128).

Lines 253 – 256: “Furthermore, the random effect of GEI significantly affected all traits. These findings also indicate that genotypes do not show consistent behavior across environments, and this should be taken into account when selecting genotypes.” -----These inferences cannot be made from Table 2 as it is. These is need for a summarized effects table showing whether G, GxE, e (e for residual) are significant for each trait. Remember you used REML to model the ‘1st’ and ‘2nd’ equations. I need to see some threshold to declare significance. This should be fairly simple; the good thing, you already have data!

Line 266 - 271: What is a “low values of CV” that defines “good experimental precision” ? Please explain briefly and provide some reference (citation). Also, please compare the same trait in the three environments, not between one trait in one environment and another trait in a different environment.

Line 271: “…denoting that all experimental conditions were equally reliable.”----- Equally variable? How was this tested? CV for Yield is 71.68 in Environment A, but it is 52.07 in Environment C. Also, the means are 1.90 in A, but 3.98 in C. The ranges look a lot closer between A and C, but not between these two environments and B. What does this imply?

Lines 273 – 283 with reference to Figure 2: The fit lines for the correlation plots for LF-SS; TS-SS, and WS-SS appear positive, while the correlation plot for YP-SS appears to be negative. Are the nature of the corresponding correlation values correct? Also see lines 321 – 323.

Line 312 … ‘accounted for’, not ‘account for’

Lines 337 – 339: ‘The AEC abscissa is the straight line that passes through the biplot origin and the “average environment”, and the AEC ordinate is the line perpendicular to the abscissa. The projections of the genotype on the abscissa represent the main genetic effects and therefore rank the genotypes in relation to their mean performance’. ------This extra information is not necessary here. Please reference the relevant biplot figure and let the reader glean this information in the corresponding caption for that figure.

Lines 342 – 345: Which figure is being referenced here? Figures 5a and 5b? Some numbers are hardly legible.

Lines 378 – 394: It is my opinion that if this section is not a procedure that enables the achievement of the main objective, it does not belong in this manuscript. Thirty lines have been tested in different environments for variations in traits associated with fruit anatomy and quality. The germplasm have been ranked using the procedures described in the text. It is not clear how testing reproductive compatibility (self and outcrosses) and the results of such tests have added value or relevance to the tenor of this manuscript. It is confusing at best.

Figure 2: The direction of the fit line seem to conflict with the sign assigned to the corresponding r: LF:SS, TS:SS, WS:SS, line shows positive orientation, but the corresponding r values are negative. YP:SS, line appears positive, yet r is negative (?!). Please check to see this is correct.

Discussion

Line 407 - 411: N=180 is being introduced here for the first time? I want to see this description in the materials and method, under plant materials; it tells us how the material in the present study was arrived at.

Line 411: was sampling random or not for the 100 ‘individuals’?

Line 414: ….reevaluated these full-sibs. Add ‘30’ before ‘full-sibs’.

Line 415: replace ‘to confirm that’ with ‘to determine if’.

Lines 425 – 426: Add citation at the end of this sentence.

Line 434: As stated in an earlier comment, what do you consider as low CV? I suggest you also incorporate what the CVs tell about the variation of the traits between the environments.

Lines 438: which _based on. Add ‘was’

Lines 442 – 444: “In the original population (N= 100), high heritabilities for the same fruit traits (except Yield) were estimated, varying from 0.59 (WP) to 0.82 (SS).”------Are these results shown anywhere in the manuscript, or published elsewhere? And what is ‘original’ population?! N=180, 100, 30 or 6? I suggest you purge the term ‘original’ here and elsewhere with reference to population. Also, remember you are working with a small sample of 30.

Lines 444 – 446: any citations?

Lines 457 – 461: see comments for lines 442 – 444, and elsewhere.

Line 463: What are ‘dissociations’ in this context? Define it or replace with a more appropriate word.

Lines 470 – 473: What does the genetic correlation between traits tell you about the variability in the phenotypes? How is this important to breeders? Please add a brief discussion on this.

Line 505: As posted earlier, what is the substantive difference between WP and YP?!

Lines 518 – 537: Decide if this study was a reproductive compatibility test or not. Otherwise, I do not see the need for these conclusions here. See comments elsewhere.

Reviewer #2: The manuscript presented the phenotypic evaluation of 30 full-sibs in a different environment. The objective is clear, and the methods and results are well presented. Statistical models were used to comprehend the potential and performance of each line and selections were made as reported in the manuscript. However, a whole set of the populations should have been used in the evaluation. The genotype data that have been already generated (Pereira et al., 2017) could have been used to conduct a detailed mapping and genetic characterization of the trait, in addition to genotype by environment interaction studies. Such effort would make the manuscript much stronger and of interest to wider readers of the journal.

6. PLOS authors have the option to publish the peer review history of their article (what does this mean?). If published, this will include your full peer review and any attached files.

Reviewer #1: No

Reviewer #2: No

---

## [Author Response · Author response to Decision Letter 0]

13 Dec 2019

A file with the Responses to reviewers and editor comments was uploaded.

---

## [Decision Letter · Decision Letter 1]

26 Mar 2020

PONE-D-19-24935R1

Improving yield and fruit quality traits in sweet passion fruit: evidence for genotype by environment interaction and selection of promising genotypes

PLOS ONE

Dear Professor Vieira,

Thank you for submitting your manuscript to PLOS ONE. After careful consideration, we feel that it has merit but does not fully meet PLOS ONE’s publication criteria as it currently stands. Therefore, we invite you to submit a revised version of the manuscript that addresses the points raised during the review process.

Dear authors, the manuscript PONE-D-19-24935R1 is the result of great work and has merit for publication. However, there are several inconsistencies in the statistical procedures that must be corrected before I consider it for publication. Another important point, English has several technical errors and needs to be reviewed by a specialized company. For this reason, I invite the authors to respond point-by-point to my comments below and that of Reviewer 3 in a reply letter. Check all changes made to the text in red. It is not necessary to use change tracking. After that, I myself will review the manuscript and provide a Decision.

We would appreciate receiving your revised manuscript by May 10 2020 11:59PM. To enhance the reproducibility of your results, we recommend that if applicable you deposit your laboratory protocols in protocols.io, where a protocol can be assigned its own identifier (DOI) such that it can be cited independently in the future. For instructions see: http://journals.plos.org/plosone/s/submission-guidelines#loc-laboratory-protocols

We look forward to receiving your revised manuscript.

Kind regards,

Paulo Eduardo Teodoro, Dr.

Academic Editor

PLOS ONE

Additional Editor Comments (if provided):

Dear authors, the manuscript PONE-D-19-24935R1 is the result of great work and has merit for publication. However, there are several inconsistencies in the statistical procedures that must be corrected before I consider it for publication. Another important point, English has several technical errors and needs to be reviewed by a specialized company. For this reason, I invite the authors to respond point-by-point to my comments below and that of Reviewer 3 in a reply letter. Check all changes made to the text in red. It is not necessary to use change tracking.

1. The authors used Linear mixed-model analysis using the ASReml-R package for individual and joint analyzes of variance. Therefore, it is necessary to write equations 1 and 2 in the form of matrix notation;

2. The authors used an identity matrix in these analyzes (matrix I). However, it is possible to estimate the kinship matrix for the assessed population;

3. Move Table 1 for supplementary material. In its place, provide a new Table containing the results of the LRT test for each trace in each environment and for each trace considering the joint analysis;

4. Include in Table 2 the genetic parameters for each trait considering the joint analysis;

5. lines 203 and 204: "genotypic correlations among traits were calculated for adjusted means, such as the Pearson coefficient, using program R". What did the authors mean by that? How were genotypic correlations estimated? Were they estimated from the BLUPs obtained? Further detail this procedure.

6. The authors used the GGE biplot method to investigate the patterns of the GxE interaction. Although I really like this method and use it in several of my papers, it is not suitable for this case. The GGE biplot method is indicated for a large number of environments, which is not the case in this study. In the case of a few environments (n = 3) in this manuscript, I suggest the authors use the HMPRGV of Resende (2007) method. As the authors are working with mixed models and have unbalances, this method would be ideal to demonstrate the genotypes that have greater adaptability and stability.

7. In the face of such corrections, the authors will need to modify the end of the Introduction. In addition, the Results will be substantially modified and, consequently, the Discussion.

Reviewers' comments:

Reviewer's Responses to Questions

**Comments to the Author**

1. If the authors have adequately addressed your comments raised in a previous round of review and you feel that this manuscript is now acceptable for publication, you may indicate that here to bypass the “Comments to the Author” section, enter your conflict of interest statement in the “Confidential to Editor” section, and submit your "Accept" recommendation.

Reviewer #3: All comments have been addressed

2. Is the manuscript technically sound, and do the data support the conclusions?

Reviewer #3: Yes

3. Has the statistical analysis been performed appropriately and rigorously? 

Reviewer #3: Yes

4. Have the authors made all data underlying the findings in their manuscript fully available?

Reviewer #3: Yes

5. Is the manuscript presented in an intelligible fashion and written in standard English?

Reviewer #3: Yes

6. Review Comments to the Author

Reviewer #3: The manuscript shows important and novelty information about phenotypic evaluation of 30 sweet passion fruit full-sibs in three a different environment, and contributes with relevant results for the scientific community and researchers interested in this study. the objective of this study was reevaluate 30 genotypes previously selected for fruit quality from a 100 full-sib sweet passion fruit progeny in three environments, with a view to estimate the heritability and genetic correlations, and investigating the GEI and response to selection for nine fruit traits (weight, diameter and length of the fruit; thickness and weight of skin; weight and yield of fruit pulp; soluble solids, and yield). The paper is acceptable in terms of methods and procedures.

Minor corrections

Line 45, 106, 120, 154, 222, and 253 - Indent the first line of a paragraph

Page 16 Line 37, Page 17 line 81, page 20 line 149, page 21 line 183, page 22 line 207 - Indent the first line of a paragraph

Page 15 line 28: What is reason for such a high coefficient of variation for yield (Environment A = 71,68, Environment B = 64,38 and Environment C = 52,07)?

Table 1. I suggest identifying abbreviation and acronyms at the bottom of the table.

Why didn't you make any mention of the parents in results and discussion since in this study, you examined 32 genotypes (page 5, line 106), consisting of a sample (n = 30) of full-sib progeny of sweet passion fruit and the two parents. Would it be possible to insert information in table 1 and table 2 using the parents as controls?

7. PLOS authors have the option to publish the peer review history of their article (what does this mean?). If published, this will include your full peer review and any attached files.

Reviewer #3: No

---

## [Author Response · Author response to Decision Letter 1]

21 Apr 2020

Please see the Cover letter, the responses to the editor and the reviewer (#3) are embedded in the letter.

---

## [Editor Report · Decision Letter 2]

23 Apr 2020

Improving yield and fruit quality traits in sweet passion fruit: evidence for genotype by environment interaction and selection of promising genotypes

PONE-D-19-24935R2

Dear Dr. Vieira,

We are pleased to inform you that your manuscript has been judged scientifically suitable for publication and will be formally accepted for publication once it complies with all outstanding technical requirements.

With kind regards,

Paulo Eduardo Teodoro, Dr.

Academic Editor

PLOS ONE

Additional Editor Comments (optional):

Dear authors, a significant minority was made in this manuscript. Therefore, I recommend its publication in Plos One.
---

## [Editor Report · Acceptance letter]

30 Apr 2020

PONE-D-19-24935R2 

Improving yield and fruit quality traits in sweet passion fruit: evidence for genotype by environment interaction and selection of promising genotypes 

Dear Dr. Vieira:

I am pleased to inform you that your manuscript has been deemed suitable for publication in PLOS ONE. Congratulations! Your manuscript is now with our production department. 

With kind regards,

on behalf of

Professor Paulo Eduardo Teodoro 

Academic Editor

PLOS ONE